# Two decades of in-situ temperature measurements in the upper troposphere and lowermost stratosphere from IAGOS long-term routine observation

Florian Berkes[1], Patrick Neis[1], Martin G. Schultz[1], Ulrich Bundke[1], Susanne Rohs[1], Herman G.J. Smit[1], Andreas Wahner[1], Paul Konopka[2], Damien Boulanger[3], Philippe Nédélec[3], Valerie Thouret[3], and Andreas Petzold[1]

[1]Forschungszentrum Jülich, IEK-8, Jülich, Germany
[2]Forschungszentrum Jülich, IEK-7, Jülich, Germany
[3]Laboratoire d'Aérologie, CNRS and Université de Toulouse, Toulouse, France

Correspondence to: Florian Berkes (f.berkes@fz-juelich.de)

**Abstract.** Despite several studies on temperature trends in the tropopause region, a comprehensive understanding of the evolution of temperatures in this climate-sensitive region of the atmosphere remains elusive. Here we present a unique global-scale, long-term data set of high-resolution in-situ temperature data measured aboard passenger aircraft within the European Research Infrastructure IAGOS (In-service Aircraft for a Global Observing System, www.iagos.org). This data set is used to investigate temperature trends within the global upper troposphere and lowermost stratosphere (UTLS) for the period 1995 to 2012 in different geographical regions and vertical layers of the UTLS. The largest amount of observations is available over the North Atlantic. Here, a neutral temperature trend is found within the lowermost stratosphere. This contradicts the temperature trend in the European Centre for Medium Range Weather Forecast (ECMWF) ERA-Interim reanalysis, where a significant (95% confidence) temperature increase of +0.56 K/decade is found. Differences between trends derived from observations and reanalysis data can be traced back to changes in the temperature difference between observation and model data over the studied period. This study underpins the value of the IAGOS temperature observations as anchor point for the evaluation of reanalyses and its suitability for independent trend analyses.

## 1 Introduction

Temperature changes in the lower stratosphere (~50 hPa) obtained from radiosondes and satellite retrievals show cooling of about 0.5 K/decade over much of the globe during the period from 1979 to 1995. Since 1995, the cooling turned into a neutral trend with larger increase (not significant) over the Antarctic region than over the tropics (Randel et al., 2009; Blunden et al., 2014; Seidel et al., 2016). The robustness of the temperature trends in the lower stratosphere derived from radiosondes (since 1958), and from satellites (since 1979) suffers from instrumental uncertainties such as sensor changes, drifts, etc., implying large uncertainties in the trend estimates (Simmons et al., 2014). In recent years, several studies

assessed the uncertainty of temperature trends in the lower stratosphere, and the impact on changing trends of radiatively active constituents (such as ozone) or atmospheric dynamics (e.g. Fueglistaler et al., 2014; Seidel et al., 2016).

Temperature trends in the upper troposphere - lowermost stratosphere (UTLS) are even more uncertain due to insufficient regional coverage of in-situ observations. Most studies of UTLS temperature trends are based on the global radiosonde network, but most of these data are from the Northern Hemisphere mid-latitudes (70% of radiosonde launches occurred between 30° and 60°N). Furthermore, these observations suffer from time-varying biases, which cannot capture the large variability of the UTLS and inhomogeneity due to changes in instrumentation (Bencherif et al., 2006; Seidel and Randel, 2006; Xu and Powell, 2010). Satellite observations cover the spatial scale, but are limited by their coarse vertical resolution, especially in the UTLS region. A well-suited data source for temperature profiles is the relatively new global positioning system radio occultation technique (GPSRO) (Kursinski et al., 1997; Wickert et al., 2001). The reliability of these measurements within the UTLS region has been demonstrated with trend analyses of GPSRO temperature (Steiner et al., 2009), or GPSRO derived trends of the thermal tropopause temperature and tropopause height (Schmidt et al., 2010; Rieckh et al., 2014; Khandu et al., 2016). Ho et al. (2017) demonstrated the usefulness of the GPSRO measurements to correct the temperature bias of radiosondes with different sensor types in the lowermost stratosphere (LMS), which is an important task to reduce the temperature uncertainties. Comparison of tropospheric temperature trends derived from homogenized satellite data sets and model simulations find more consistency, but require long-term time series (>17 years) before a robust trend arises from internal climate variability (Santer et al., 2011).

Since the radiative forcing from greenhouse gases, including water vapor, is sensitive to changes in the mid-troposphere and the UTLS (Solomon et al., 2010; Riese et al., 2012), this region is extremely important for climate change and for controlling dynamical processes governing stratosphere-troposphere exchange (Gettelman et al., 2011). Furthermore, the variability and changes of the temperature in the UTLS play an important role in regulating the exchange of water vapor, ozone, and other trace gases between the troposphere and the stratosphere.

Continuous in-situ observations of these properties in the UTLS region can only be conducted with satellite and aircraft measurements over a large spatial region. Automated aircraft temperature observations are collected, along with an increasing amount of humidity data through the World Meteorological Organization (WMO) Aircraft Meteorological DAta Relay (AMDAR) program (WMO, 2014; Petersen, 2016). Petersen (2016) showed that at flight level the errors of temperature and wind in the 3-48 h forecast were reduced by nearly 50% when assimilating data from passenger aircraft. However, Ballish and Kumar (2008) and Drüeet al. (2008) identified that the AMDAR aircraft temperature is strongly affected by a warm bias, which can fluctuate by altitude, aircraft type and phase of flight, while the reason for this bias is not fully understood (Ingleby et al., 2016).

Previous studies already used the passenger aircraft measurements from MOZAIC (Measurement of Ozone and Water Vapour on Airbus In-service Aircraft, Marenco et al., 1998) temperature measurements for inter-comparison with GPSRO measurements and ECMWF (European Centre for Medium Range Weather Forecast) analyses. Bortz et al. (2006) analyzed MOZAIC temperature measurements at cruise altitude from 1994 to 2003 within the tropical region. They found no significant temperature increase within the upper tropical troposphere. The authors concluded that the temperature measurements are well representative to be used in inter-comparison studies with satellite (MSU) and radiosonde measurements within this region, but at this time the data record was too short for trend estimates. Heise et al. (2008) compared around 2700 MOZAIC in-situ temperature profiles against profiles from GPSRO and analysis from ECMWF between 2001 and 2006. They concluded that MOZAIC in-situ temperature had no bias against the ECMWF temperature above 300 hPa, whereas GPSRO showed a cold bias of -0.9 K compared to the MOZAIC temperature. Since 2001 GPSRO data are assimilated in numerical weather prediction models and reanalysis products. Schmidt et al. (2010) characterized the tropopause inversion layer in the Northern Hemisphere with temperature profiles from in-situ measurements and from the model for the period from 2001 to 2009 and concluded that the cold point at the tropopause agreed well.

In this study, IAGOS (In-service Aircraft for a Global Observing System) temperature observations are analyzed, which are available for almost two decades since 1994. The geographical coverage of the measurements is shown in section 2, where the reliability of the IAGOS observations and the data selection are discussed. The UTLS temperature distribution and derived temperature trends are presented in section 3, and their robustness and suitability for the evaluation of global scale reanalyses with the example of ERA-Interim are discussed in section 4. Due to varying geographical coverage of these data they cannot provide a full global assessment yet, but as we will show in our study, they can serve as anchor point for trend analyses and evaluation of reanalysis at least over the extratropics. This conclusion is included in section 5.

## 2 Data selection and methods

### 2.1 The IAGOS European Research Infrastructure

Since 1994, IAGOS in-situ observations of essential climate variables (temperature, water vapour, and ozone) in the UTLS are provided on a global scale by the European Research Infrastructure IAGOS (Petzold et al., 2015). IAGOS builds on the former EU framework projects MOZAIC and CARIBIC (Civil Aircraft for the Regular Investigation of the Atmosphere Based on an Instrument Container, Brenninkmeijer et al., 2007).

Currently (2017) up to ten passenger aircraft from various international airlines are equipped with scientific instruments to monitor the meteorological state (temperature, water vapor and wind) and atmospheric chemical composition. Besides the measurements of temperature and (relative) humidity, the IAGOS-CORE Package 1 includes instruments to measure ozone and carbon monoxide. Ozone is measured by UV absorption (Thermo Scientific for Model 49), and CO by IR-absorption

(Thermo Scientific for Model 48 Trace Level). Both instruments are regularly calibrated before and after deployment and the overall uncertainty for ozone is 2 ppb ±2% and for CO 5 ppb ±5% (Thouret et al., 1998; Nédélec et al., 2015). Additionally, positions, pressure, ambient temperature (measured from the aircraft), aircraft speed, wind speed and wind direction are provided by the A330 and A340 avionic system, respectively (details are given by Petzold et al. (2015)). For completeness, several additionally parameters ($NO_y$, $NO$, $NO_2$) are measured or will be soon measured ($CH_4$, $CO_2$) from IAGOS Packages 2 which are described by Petzold et al. (2015).

## 2.2 Temperature measurements and evaluations

On IAGOS aircraft, temperature is measured in-situ with a compact airborne sensing device AD-FS2 (Aerodata, Braunschweig Germany), which is installed in an appropriate aeronautic housing (Helten et al., 1998). From 1994 to 2009 a platinum resistance sensor (Pt100) was attached at the humidity sensing device. After a design change from MOZAIC to IAGOS this sensor is now directly at the humidity sensor (Figure 1). Inter-comparison between both systems showed a temperature deviation of less than 0.1 K in the calibration chamber, which is below the sensor uncertainty. The temperature is measured with an uncertainty of ±0.25 K by a microprocessor controlled transmitter unit (Model HMT 333, Vaisala, Finland), which passes the signal to the data acquisition system of IAGOS Package 1 instrument, where it is recorded with a time resolution of 4 s (Nédélec et al., 2015; Petzold et al., 2015). Pre- and post-deployment calibrations from the laboratory are used to evaluate the temperature signal and to ensure the quality of the temperature measurement (Helten et al. 1998, Neis et al. 2015). Typical deployment phases are in the range of 2 to 3 months. Accounting for corrections of adiabatic compression at the inlet part of the housing (Stickney et al., 1994; Moninger et al., 2003), the overall uncertainty of the ambient air temperature is ±0.5 K. More details are given in the standard operation procedure (SOP) of the IAGOS capacitive Hygrometer (ICH), available at www.iagos.org. For the purpose of this analysis the dataset was reduced to 1-minute averages.

To ensure the reliability of the IAGOS temperature observations, we make use of temperature measurements from the AIRTOSS-ICE aircraft campaign (Aircraft Towed Sensor Shuttle – Inhomogeneous Cirrus Experiment) which focused on mid-latitude cirrus clouds (Neis et al., 2015). During this campaign, the IAGOS temperature instrument was installed on the research aircraft (Learjet 35A), and provides the opportunity to compare both temperature measurements. The aircraft temperature measurement was made with Pt-100 thermistor mounted in the same type of Rosemount as for the IAGOS temperature measurements. The temperature sensor of the research aircraft has been calibrated regularly with an uncertainty of about 0.5 to 1.0 K.  Figure 2 shows the temperature correlation and the temperature bias ($\Delta T$) at pressures below 400 hPa during seven flights. The general behavior between both temperature measurements agreed well along the flight tracks. The mean deviation is $\Delta T = -0.3$ K, with a pressure dependency to a smaller temperature bias towards 200 hPa. The temperature correlation is high and the temperature bias is smaller than the overall uncertainty of 0.5 K (sensor and adiabatic

compression correction), which demonstrates the capability of the IAGOS temperature sensor to measure the ambient temperature at cruise altitude very precisely.

The temperature measured from the aircraft ($T_{AC}$) is based on total air temperature (TAT) designed for subsonic aircraft
(Goodrich Corporation, formerly Rosemount Aerospace). The total air temperature is defined as the ambient air temperature plus the temperature increase due to adiabatic compression in the Rosemount housing. Typically three TAT-sensors (platinum resistance sensor) are installed at the nose region of the aircraft, but in general only one is used for the pilots and stored for IAGOS. The other two sensors are used to monitor the differences between all TAT-sensors. In general, the airline follow the AMDAR quality recommendations (WMO, 2003), and the TAT-sensors are regular checked by visible inspection.
An exchange of one TAT-sensor is performed, if it differs more than 3°C from the other two TAT-sensors.

## 2.3 Spatial and temporal data coverage

In this study all temperature measurements from the IAGOS-CORE flights at cruise altitude (p <350 hPa) from January 1995 to December 2012 are used. More recent measurements are not yet validated and therefore not included in this study, therefore most of the IAGOS temperature observations rely on the former instrument design. Following previous IAGOS or
MOZAIC analyses (Thouret et al., 2006; Dyroff et al., 2015; Thomas et al., 2015; Stratmann et al., 2016), we divide the data into 14 geographical regions (Figure 3 and Table 1). Seasonal and regional differences of the temperature behavior can then be linked to the different dynamical patterns. The largest amount of measurements are obtained in the mid-latitudes (North America, North Atlantic, Europe and Central Asia), whereas the amount of measurements in the northern latitudes (North Canada, Greenland, Scandinavia, North Asia) and the tropical regions (Middle America, Tropical Atlantic, North Africa,
Tropical Asia, South America, South Africa) is much smaller and does not provide a continuous coverage over the presented period. The data coverage for each region over the analyzed period is shown in Figure 4.

Changes of radiativly active chemical constituents (e.g. ozone and water vapor) within the LMS and the upper troposphere (UT) have different impacts on the ambient temperature in these layers. In general the tropopause layer (TPL) separates the
stable stratified LMS and the unstable UT. In the present study, the pressure of the thermal tropopause ($p_{TPHWMO}$) is derived from the ERA-Interim (see below), and it is used to determine the position of the aircraft relative to this layer, and to distinguish if the aircraft flew within the UT, in the TPL or the LMS. This is achieved using the following criteria:

LMS : $p < p_{TPHWMO} - 15$ hPa, which is limited by the maximum cruise altitude (p ~ 190 hPa)
TPL: $p = p_{TPHWMO} \pm 15$ hPa
UT: $p > p_{TPHWMO} + 15$ hPa, limited to 350 hPa

A comparable definition has been used by Thouret et al. (2006), where the pressure of the 2-pvu-surface was used to define the dynamical tropopause layer with a vertical depth of ±15 hPa. The thermal tropopause is valid within all latitude bands, whereas the dynamical tropopause cannot be used in tropical regions, because the Coriolis parameter is zero, therefore potential vorticity (PV) goes to zero and the 2-pvu surface is not defined (Boothe and Homeyer, 2017).

Within the IAGOS project, measurements of ozone (since 1994) and carbon monoxide (since 2002) are available and are used to justify our layer classification scheme. Figure 5 shows the cumulative distribution of ozone, carbon monoxide and potential vorticity (PV). All distributions demonstrate a clear separation between the three layers based on the thermal tropopause. Median ozone mixing rations in the UT are around 60 ppb, median CO mixing ratios are 90 ppb, and median PV is 0.5 PVU, respectively. Within the LMS, the median values for $O_3$ (310 ppb), for CO (40 ppb) and for PV (7 PVU) are consistent to previous studies (Pan et al., 2004; Kunz et al., 2008; Brioude et al., 2009; Schmidt et al., 2010).

## 2.4 Meteorological reanalysis

ERA-Interim covers the period from 1979 until present, assimilating observational data from various satellites, radiosondes, buoys, commercial aircraft and others (Dee et al., 2011; Simmons et al., 2014). Note that the IAGOS temperature observations are not assimilated in any numerical weather prediction model or reanalysis product, which makes it unique for model evaluation. For this study, the 6-hourly output from ERA-I (0.75° x 0.75°) were interpolated onto a 1° x 1° horizontal grid, and on 60 vertical levels of constant pressure and potential temperature (Kunz et al., 2014). Additionally, the variables of the potential vorticity (PV), and the pressure of the thermal tropopause ($p_{TPHWMO}$) based on the WMO criteria were calculated (WMO, 1957; Reichler et al., 2003). The ERA-I data were linearly interpolated (longitude, latitude, pressure, time) onto each flight track with 4 s resolution as described by Kunz et al. (2014). As for IAGOS temperature observations the ERA-I dataset was reduced to 1-minute averages.

## 3 Results

### 3.1 IAGOS temperature measurements over different regions

Approximately 69% of the IAGOS temperature measurements in the UTLS were obtained in the mid-latitude band between 30° to 60°N. Therefore, we show our detailed analysis within all three layers of the UTLS only over the North Atlantic region between January 1995 and December 2012 and provide additional material as supplement. Figure 6 shows monthly median temperature of the IAGOS observations in the LMS, TPL and UT for the period from 1995 to 2012 over the North Atlantic region. All layers show a seasonal temperature variation of 5 to 10 K in each year. The warmest temperatures are observed within the LMS and the coldest (as expected) within the TPL. Between 2006 and 2008, the wintertime temperature

minima at the TPL and the UT are almost 3 to 4 K warmer compared to the other years. These enhanced temperatures can also be observed over Europe and Central Asia within these layers. All other regions in the extratropics show mostly comparable time series in amplitude and seasonal variations to the North Atlantic region, if enough measurements were available for each vertical layer. Throughout all regions in the extratropics, the summer / winter months (JJA / DJF) are always the warmest / coldest within all three layers (see tables in the supplement). Over the tropical regions, no measurements are available in the LMS and TPL, because the TPL is mostly above cruise altitude. In the UT, the temperature is mostly constant throughout the year for each tropical region respectively.

## 3.2 Temperature anomalies and trends

The temperature trends are derived from using a robust regression analysis over the full period of deseasonalized temperature observations. Eighteen-year monthly averages (e.g. mean of all January values, mean of all February values, etc.) were subtracted from the time series of each layer in all regions. The temperature trends exhibit no significant non-linear contributions. We ensured that the data is homoscedastic, and it is nearly normally distributed (median and mean values are very close, see table in supplements). Additional, we checked that the data is not auto-correlated. Temperature trends are reported only if at least 90% of all months have at least 200 data points each (Figure 4). The Mann-Kendall-test is used to identify the trend significance (Mann, 1945; Kendall, 1975; Gilbert, 1987). The 90% threshold was chosen because the Mann-Kendall significance of the trend analysis did not change, when 10% of the data were randomly excluded from the trend calculation. The robustness of temperature trends was tested by skipping the first or last year of the 18-year period. Within all layers and all regions each trend keeps the same sign and the trend values varied within the standard error. The only exception was the upper troposphere over North America where the temperature trend changed from slight positive trend (18 years) to neutral when the final year was removed.

Figure 7 shows the time series of temperature anomalies and linear trend lines over the North Atlantic region using the IAGOS observations. The monthly temperature anomalies in each layer vary by ±3 K (strongest within the UT). Temperature trends are +0.22 (±0.20) K/dec in the UT, +0.25 (±0.16) K/dec in the TPL and -0.05 (±0.17) K/dec in the LMS. None of the trends are significant at the 95% level. In the UT, the amplitudes of the smoothed time series are larger and decrease towards the upper levels. They show two warmer phases and two colder phases during the analyzed period.

Figure 8 and Table 3 summarize the temperature trends for different regions in the different layers. Within the LMS, significant cooling is observed over Greenland (-1.39 (±0.29) K/dec) and North America (-0.71 (±0.21) K/dec). The smaller trend over Europe (-0.53 (±0.20) K/dec) is not significant. The other regions don't have enough data for a meaningful trend

analysis in the LMS. Within the TPL, there are only three regions with sufficient data coverage, and no significant trend is detected, although there is a weak indication for a small warming over North America and the North Atlantic and for a small cooling over Europe. In the UT, significant cooling is found over North America (-1.08 ($\pm$0.18) K/dec), while temperatures over Europe (-0.59 ($\pm$0.1) K/dec) decrease not significant. Over the North Atlantic (+0.22 ($\pm$0.20) K/dec) and over Central Asia (+0.32 ($\pm$0.33) K/dec), temperatures increase non-significantly, respectively. Over Tropical Asia (-0.54 ($\pm$0.04) K/dec) temperature show a tendency to decrease, but without significance at the 95% level. This result is somehow puzzling, because it is assumed that the temperature increases in the tropics in the UT, equivalent to the surface temperature (Khandu et al., 2016). One reason could be related to the tropopause definition, but $O_3$ (40 to 80 ppb) and PV (always below 1 PVU) indicate that the selected air mass is tropospheric. Another reason could be related to a higher variability of the temperature due to large scale influence or simply that the data coverage is still too poor in this region, which might lead to a higher variability of the local temperatures which then mask the temperature trend.

## 4 Discussion

### 4.1 Inter-comparison with ERA-Interim reanalysis

ERA-interim reanalysis is widely used for intercomparions and is currently the latest global atmospheric reanalysis product provided by ECMWF (Dee et al., 2011; Fujiwara et al., 2017). Simmons et al. (2014) indicated that there is a problem with the temperature near the tropopause in the tropics and extratropics. Here, we demonstrate the use of IAGOS observations to evaluate the ERA-I reanalysis within the UTLS region.

Figure 9 shows the inter-comparison between the temperature of the IAGOS observations and the ERA-I reanalysis of all 1-minute mean data over the North Atlantic region at cruise altitude (below 350 hPa). The good agreement between both temperatures is reflected by the high correlation coefficient ($R^2 = 0.97$) and the slope of 0.94 from the regression fit. The bias between the temperature of the IAGOS measurements and ERA-I is -0.02 K over the entire period. The absolute bias is 0.81 ($\pm$0.72) K and 93% of the data is close to the 1:1-line ($\pm$2K). The spread of the correlation corresponds to a larger variability of the IAGOS temperature measurements when the aircraft flew through clouds, or from the interpolation of the gridded ERA-I temperature to the aircraft position. Since 2011, IAGOS measurements with a backscatter cloud probe are available and these measurements will be used in the future to distinguish between clear sky and in-cloud measurements to further reduce the uncertainties. The good agreement between the 1-minute mean temperature data from ERA-I and the IAGOS observation in the other regions is documented in Table 2, when $R^2$ is always larger than 0.97, the slope of each regression fit is between 0.94 and 0.98, and the absolute bias varies between 0.60 to 0.98 K with a standard deviation of 0.56 to 0.79 K.

## 4.2 Anomalies and trends using ERA-I

The ERA-I temperature time series and temperature anomalies agree with the IAGOS measurements mostly in phase and amplitude within all vertical layers over the North Atlantic region (Figure 6 and Figure 7). Larger deviations can be found at the beginning of the time series in the mid-1990s, where the IAGOS temperatures are warmer than ERA-I, and after 2008, when the opposite occurs. The temperature trends from ERA-I show a slight warming over the North Atlantic region (+0.38 (±0.18) K/dec) in the UT, and 0.46 (±0.15) K/dec in the TPL. Neither the trends in the UT nor the trend in the TPL are significant.

In the LMS, ERA-I temperatures show a significant increase of +0.56 (±0.17) K/dec over the 18 years of the study period at the 95% significance level, which is not the case for the IAGOS observations, where the temperature shows a not significant decrease of -0.05 (±0.17) K/dec over the analyzed period (see section 4.1 above). We calculated the temperature trends of the lower (5-percentile, colder temperatures) and upper (95-percentile warmer temperature) range of the data to determine the robustness of these trends within the LMS. For the 5-percentiles, the temperature trend from ERA-I remains significant (99% confidence) at +0.48 K/dec, while the trend from the IAGOS observations is still not significantly decreasing at -0.22 K/dec (87% confidence). In the upper range, the differences between the temperature trends diminish to +0.12 K/dec (53% confidence) using ERA-I and -0.07 K/dec (73% confidence) using the IAGOS observations. This shows that within the colder temperature regimes, the trend difference is significant, while for the warmest temperatures regimes, the variability is too large to obtain a significant temperature trend.

Figure 8 and Table 3 also contain the temperature trend estimates for ERA-Interim. In contrast to the IAGOS observations, a cooling tendency in the LMS can only be found over Greenland of (-0.79 (±0.29) K/dec), and over North America (-0.25 (±0.21) K/dec), but both are not significant. The strong cooling trend over Greenland results from elevated temperatures in the late 1990s. Over Europe, where IAGOS observations showed an insignificant decrease, ERA-I exhibits a very small (and not significant) warming of +0.11 (±0.19) K/dec.

Within the TPL, the temperatures trends are mostly comparable to the IAGOS observations (expect over Europe), but all trends are not significant. The best consistency between the temperature trends from IAGOS observations and ERA-I is seen for the UT layer. Except over Europe, where ERA-I shows slightly less cooling than IAGOS observations, and Central Asia, where ERA-I has a stronger warming trend.

## 4.3 Possible sources of the differing temperature trends

There is a remarkable coherence of the temperature time series and the anomalies of the observations and ERA-I in all regions and layers, but the linear temperature trends reveal a deviation between the two data sets, especially in the LMS. In

order to investigate this in more detail, Figure 10 shows the annual deviation between the IAGOS observations and ERA-I at the LMS over eight regions. From 1995 to 2002 the observations are always warmer than ERA-I within the Northern Hemisphere. The sign changes to neutral until 2008, and since 2009 the IAGOS observations are colder than ERA-I. This temporally varying temperature bias is one reason why the temperature trends are not equal between the two data sets. The differences cannot be explained by the chosen definition to determine the different layers within the UTLS region. For example, ozone observations show seasonal variations with values between 350 to 640 ppb, which are robust values for stratospheric air masses with a typical seasonality (Brioude et al., 2009). Additionally we use CO observations (from 2002 to 2008) and PV with similar results.

Furthermore, we demonstrated that temperature measurements from IAGOS agree well to quality controlled measurements aboard research aircraft (Figure 2). Therefore, we expect that the trends from IAGOS are robust, which leads to the hypothesis that ERA-I exhibits temperature biases which vary with time.

Some of the observed deviations can be explained by changes of the data assimilation sources in ERA-I, which employed several new satellite products at various times after the year 2000. Simmons et al. (2014) showed that the source of input data (e.g. inconsistent SST bias) changed in ERA-I after 2002. Since 2001, temperature profiles were assimilated from GPSRO measurements into ERA-I, which lead to a cold bias in ERA-I, where the largest effects appeared after 2006 in ERA-I, when the amount of assimilated data increased. This could lead to a warming of the lower stratosphere and a cooling at around 200 hPa (Poli et al. 2008; Poli et al. 2010).

Another source of the deviation could be expected from the increasing number of temperature measurements from the AMDAR system onboard of passenger aircraft. Ballish and Kumar (2008) showed that the temperatures from passenger aircraft are warm-biased at cruise altitude (200 to 300 hPa). This has not been accounted for ERA-I. Furthermore, the WMO reported that the amount of aircraft reports increased from 100 000 in early 2000 to more than 350 000 after 2012, and became the 3[rd] most important data source of assimilation for short term forecast in numerical weather prediction (WMO, 2014; Petersen, 2016).

In order to test the second hypothesis, Figure 7 also includes the temperature trend derived over the North Atlantic region using the aircraft temperature measurements ($T_{AC}$), which we assume to be comparable to AMDAR measurements. The temperature trend in the LMS is also positive, and in-between the neutral temperature trend using the IAGOS temperature observations and the positive temperature trend using ERA-I. This gives an indication that the assimilated aircraft measurements could indeed be a cause for bias in the ERA-I data. It is therefore planned to introduce a bias correction for AMDAR aircraft observations in the next ERA reanalysis (ERA5, personal communication D. Dee).

**5 Conclusion**

In this study, nearly two decades of in-situ IAGOS temperature measurements using passenger aircraft by the European Research Infrastructure IAGOS were presented and used to determine regional trends in the UTLS. The quality of the temperature measurements is regularly evaluated in the laboratory through pre- and post-deployment calibration, and it has been assessed by intercomparison with temperature observations from research aircraft.

UTLS temperature time series and trends are analyzed for 14 different regions within the northern latitudes, mid-latitudes and tropics, and separated into the lowermost stratosphere, the tropopause layer and the upper troposphere using the thermal tropopause from ERA-I as reference layer.

The inter-annual variability within all regions and layers is mostly consistent between the IAGOS observations and the ERA-I during the past 18 years (1995 to 2012), which is not the case for the temperature trends. For the temperature trend only regions are considered when at least 90% of all months are available. Over the North Atlantic where the largest amount of measurements is available, we found a significant (95% confidence level) positive trend (+0.56 K/dec) in ERA-I and a small negative trend (-0.05 K/dec) in the observations (not significant) in the LMS. A significant (95% confidence) negative temperature trend over Greenland and North America in the LMS for the IAGOS temperature is found. Over Europe both temperatures trends are not significant, but the temperature trend from the IAGOS observations is negative and for ERA-I positive. Within the tropopause layer we found mostly comparable trends, expect over Europe, where the sign of trends is different, but all trends are not significant. The calculated temperature trends in the UT show all the same sign for each region, whereas only a significant trend is found over North America with a large cooling rate of about -0.92 K/dec (ERA-I) and -1.08 K/dec (IAGOS) in the last 18 years.

The large deviation between the different LMS temperature trends from IAGOS observations and ERA-I data is mostly related to the temporally varying bias between both temperature time series. As we have no reason to assume an evolving bias in the IAGOS observations, we conclude that ERA-I temperatures are too cold between 1995 and 2001, and too warm after 2007. These dates roughly correspond with changes in the data streams that were used in the ERA-I data assimilation. The evolution of ERA-I temperature can be explained by additional assimilation of GPSRO data and data from the AMDAR passenger aircraft network, which have been shown to be warm-biased. These data sources play an increasing role in the ERA-I data assimilation after 2006. Our recommendation is therefore to include a bias correction for these temperature measurements in future versions of the European reanalysis and use IAGOS observations as anchor point.

The IAGOS temperature measurements highlight the need of independent global measurements with a high and long-term accuracy to quantify long-term changes especially in the UTLS region, and help to identify inconsistencies between different

datasets of observations and models. Due to the expansion of the IAGOS aircraft fleet with airlines from other continents, we are looking forward to investigate temperature changes in the UTLS over several more regions in a few years from now.

**Acknowledgements**

We gratefully acknowledge all partners for their continuous support for more than 20 years: Lufthansa, Air France, China Airlines, Cathay Pacific, Iberia, CNRS, University of Manchester, Meteo-France, Sabena Technics and enviscope GmbH, including all former coordinators of MOZAIC and IAGOS: A. Marenco, A. Volz-Thomas, and J.P. Cammas. We acknowledge ECMWF for providing the ERA-Interim data and N. Thomas for excellent programming support. Part of this project was funded by BMBF in IAGOS-D contract 01LK1301A. The IAGOS database is supported by AERIS (CNES and INSU-CNRS).

**Author contributions**

F.B. conducted the analysis of the measurements and wrote the manuscript. H.G.J.S., P.N., S.R., and U.B. were in charge of the instrument setup, calibration and processing of the measurements. A.P., M.G.S. and A.W. helped with the interpretation of the data and in the manuscript writing process. P.N., D.B. and V.T. were in charge to provide the CO and ozone measurements of MOZAIC and IAGOS.

**Competing interests**

The authors declare that they have no conflict of interests. The measurements are available free of charge on www.iagos.org.

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

Table 1: Definition of different regions covered by IAGOS flight tracks.

|  | Region | Longitude | Latitude |
|---|---|---|---|
| Northern latitudes | North Canada | 130°W - 65°W | 60°N - 80°N |
|  | Greenland | 65°W - 5°W | 60°N - 80°N |
|  | Scandinavia | 5°W - 45°E | 60°N - 80°N |
|  | North Asia | 45°E - 180°E | 60°N - 80°N |
| Mid-latitudes | North America | 130°W - 65°W | 30°N - 60°N |
|  | North Atlantic | 65°W - 5°W | 30°N - 60°N |
|  | Europe | 5°W - 45°E | 30°N - 60°N |
|  | Central Asia | 45°E - 180°E | 30°N - 60°N |
| Northern Tropics | Middle America | 130°W - 65°W | 0°N - 30°N |
|  | Tropical Atlantic | 65°W - 5°W | 0°N - 30°N |
|  | North Africa | 5°W - 45°E | 0°N - 30°N |
|  | Tropical Asia | 45°E - 180°E | 0°N - 30°N |
| Southern Tropics | South America | 65°W - 5°W | 30°S - 0°N |
|  | South Africa | 5°W - 45°E | 30°S - 0°N |

**Table 2: Statistical parameters summarized for all studied regions (see Figure 9). The slopes and intercepts are determined from the linear robust regression fit between ERA-I and IAGOS temperature. Additionally, for each region the absolute bias and its standard deviation are given.**

| | Region | Number of data points *[$10^6$] | $R^2$ | Slope | Intercept | Absolute Bias [K] | Standard Deviation [K] |
|---|---|---|---|---|---|---|---|
| Northern latitudes | North Canada | 0.23 | 0.98 | 0.96 | 9.68 | 0.73 | 0.64 |
| | Greenland | 0.50 | 0.97 | 0.96 | 9.83 | 0.81 | 0.70 |
| | Scandinavia | 0.16 | 0.97 | 0.96 | 9.24 | 0.79 | 0.68 |
| | North Asia | 0.61 | 0.97 | 0.95 | 10.34 | 0.92 | 0.77 |
| Mid-latitudes | North America | 1.42 | 0.98 | 0.95 | 10.30 | 0.79 | 0.69 |
| | North Atlantic | 3.23 | 0.97 | 0.94 | 12.26 | 0.81 | 0.72 |
| | Europe | 2.80 | 0.97 | 0.95 | 10.27 | 0.85 | 0.72 |
| | Central Asia | 1.76 | 0.98 | 0.97 | 6.35 | 0.92 | 0.79 |
| Northern Tropics | Middle America | 0.03 | 0.99 | 0.97 | 6.64 | 0.81 | 0.57 |
| | Tropical Atlantic | 0.39 | 0.97 | 0.97 | 7.29 | 0.77 | 0.63 |
| | North Africa | 0.71 | 0.98 | 0.97 | 6.53 | 0.66 | 0.61 |
| | Tropical Asia | 0.50 | 0.98 | 0.96 | 10.49 | 0.98 | 0.79 |
| Southern Tropics | South America | 0.14 | 0.98 | 0.97 | 6.62 | 0.87 | 0.65 |
| | South Africa | 0.41 | 0.99 | 0.98 | 4.74 | 0.60 | 0.56 |

**Table 3: Temperature trends of ERA-I and from the IAGOS observations within the LMS TPL and UT, where at least 90% of measurements were over the entire period (1995 to 2012) available. SE is the standard error of the temperature trend. The temperature trend is significant (Sig=1) if the p-value (consistency) is smaller than 0.06 (>94%), which was derived from the Mann-Kendall-test.**

| Region | ERA-I | | | | IAGOS | | | |
|---|---|---|---|---|---|---|---|---|
| | $\Delta T_{18yr}$ K/dec | SE K/dec | Sig. | p-value | $\Delta T_{18yr}$ K/dec | SE K/dec | Sig. | p-value |
| *LMS* | | | | | | | | |
| Greenland | -0.79 | 0.29 | 0 | 0.12 | -1.39 | 0.29 | 1 | 0.01 |
| North America | -0.25 | 0.21 | 0 | 0.46 | -0.71 | 0.21 | 1 | 0.02 |
| North Atlantic | +0.56 | 0.17 | 1 | 0.05 | -0.05 | 0.17 | 0 | 0.98 |
| Europe | +0.11 | 0.19 | 0 | 0.83 | -0.53 | 0.20 | 0 | 0.17 |
| *TPL* | | | | | | | | |
| North America | +0.29 | 0.19 | 0 | 0.19 | +0.23 | 0.20 | 0 | 0.28 |
| North Atlantic | +0.46 | 0.15 | 0 | 0.16 | +0.25 | 0.16 | 0 | 0.33 |
| Europe | +0.20 | 0.15 | 0 | 0.55 | -0.44 | 0.17 | 0 | 0.20 |
| *UT* | | | | | | | | |
| North America | -0.92 | 0.17 | 1 | 0.04 | -1.08 | 0.18 | 1 | 0.01 |
| North Atlantic | +0.38 | 0.18 | 0 | 0.20 | +0.22 | 0.20 | 0 | 0.52 |
| Europe | -0.24 | 0.14 | 0 | 0.73 | -0.59 | 0.15 | 0 | 0.26 |
| Central Asia | +0.66 | 0.33 | 0 | 0.53 | +0.32 | 0.33 | 0 | 0.85 |
| Tropical Asia | -0.58 | 0.39 | 0 | 0.72 | -0.54 | 0.04 | 0 | 0.63 |

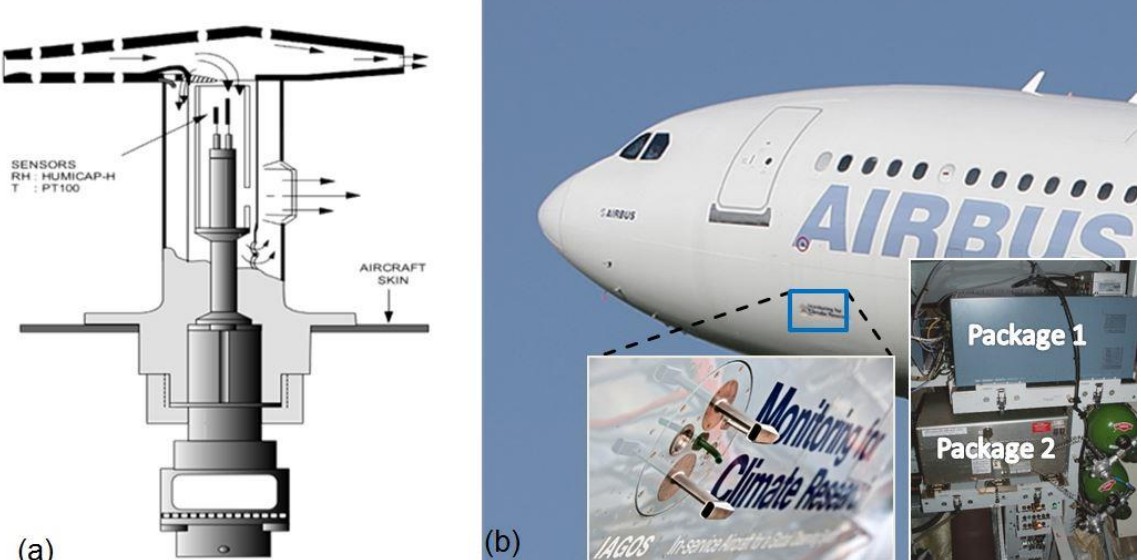

**Figure 1: a) Schematic scheme of the temperature sensor attached to the humidity sensor mounted in the Rosemount housing (Helten et al., 1998). b) Packages 1 and 2 installed aboard the AIRBUS A340-300, and the inlet plate including the Rosemount housing (photograph by courtesy of Lufthansa).**

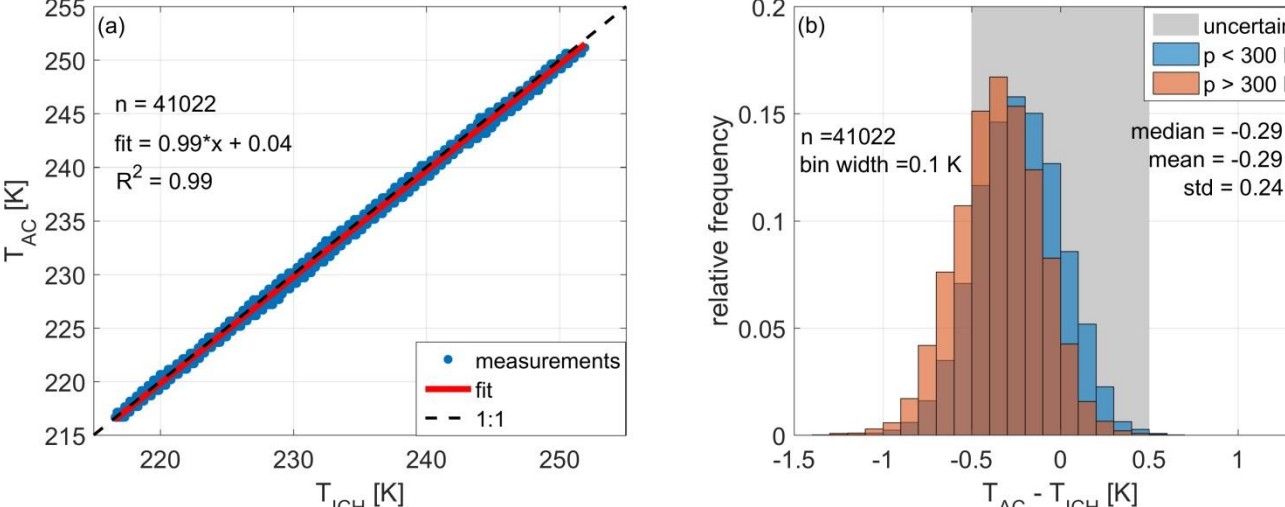

**Figure 2: (a)** Temperature correlation for all measurements below the pressure altitude 400 hPa and **(b)** Temperature bias of the IAGOS temperature instrument ($T_{ICH}$) compared to a regularly quality checked temperature sensor ($T_{AC}$) from the research aircraft for seven flights during AIRTOSS-ICE campaign in 2013. The grey area marks the total range of the total uncertainty for the IAGOS temperature observations. The temperature bias was separated for measurements between 300 and 400 hPa (orange) and 200 to 300 hPa (blue) to highlight the shift to a smaller bias at lower pressure.

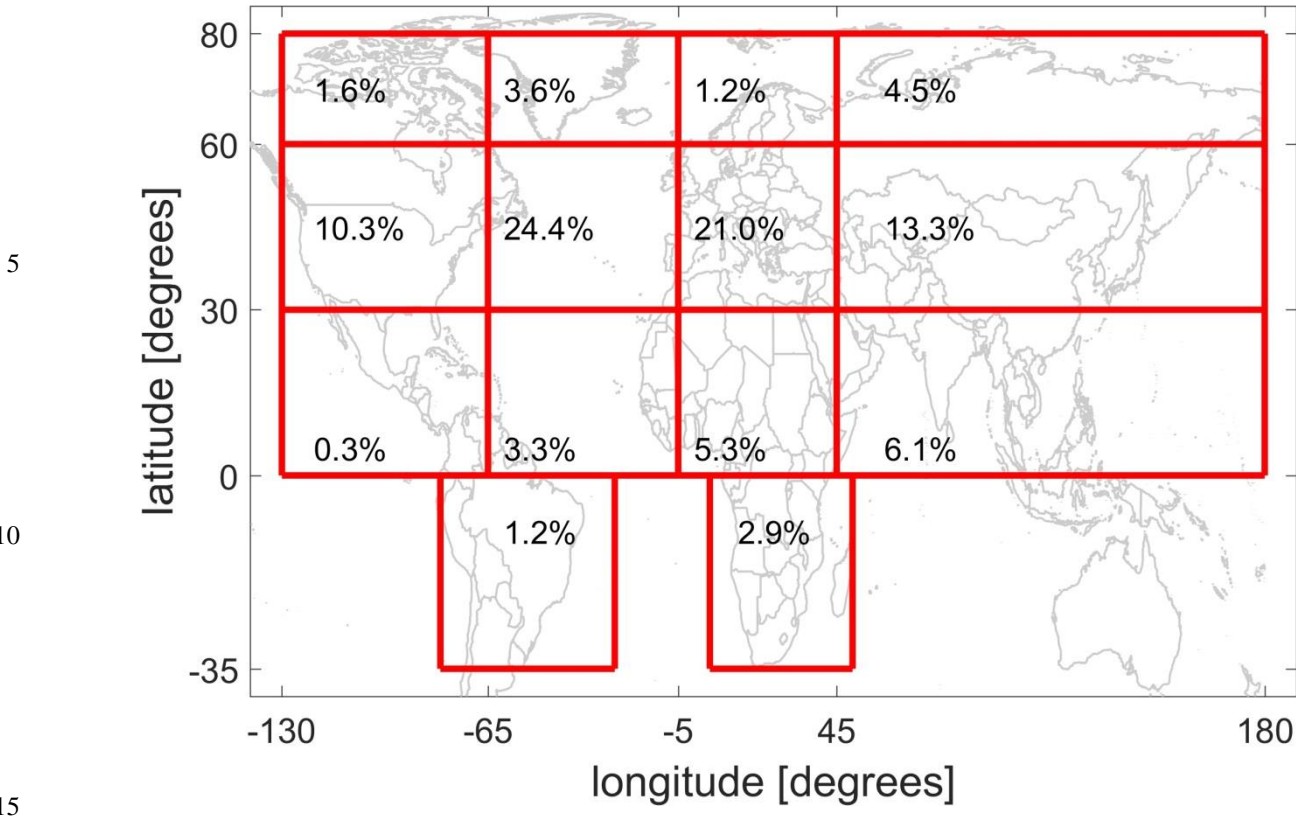

**Figure 3: Relative frequency of the IAGOS temperature observations at cruise altitude (p < 350 hPa) in different regions (Tab. 1). The total number of 1-minute averages is 14.8 million. Note, 1% includes 0.2 Million 1-minute mean data points within the ULTS region**

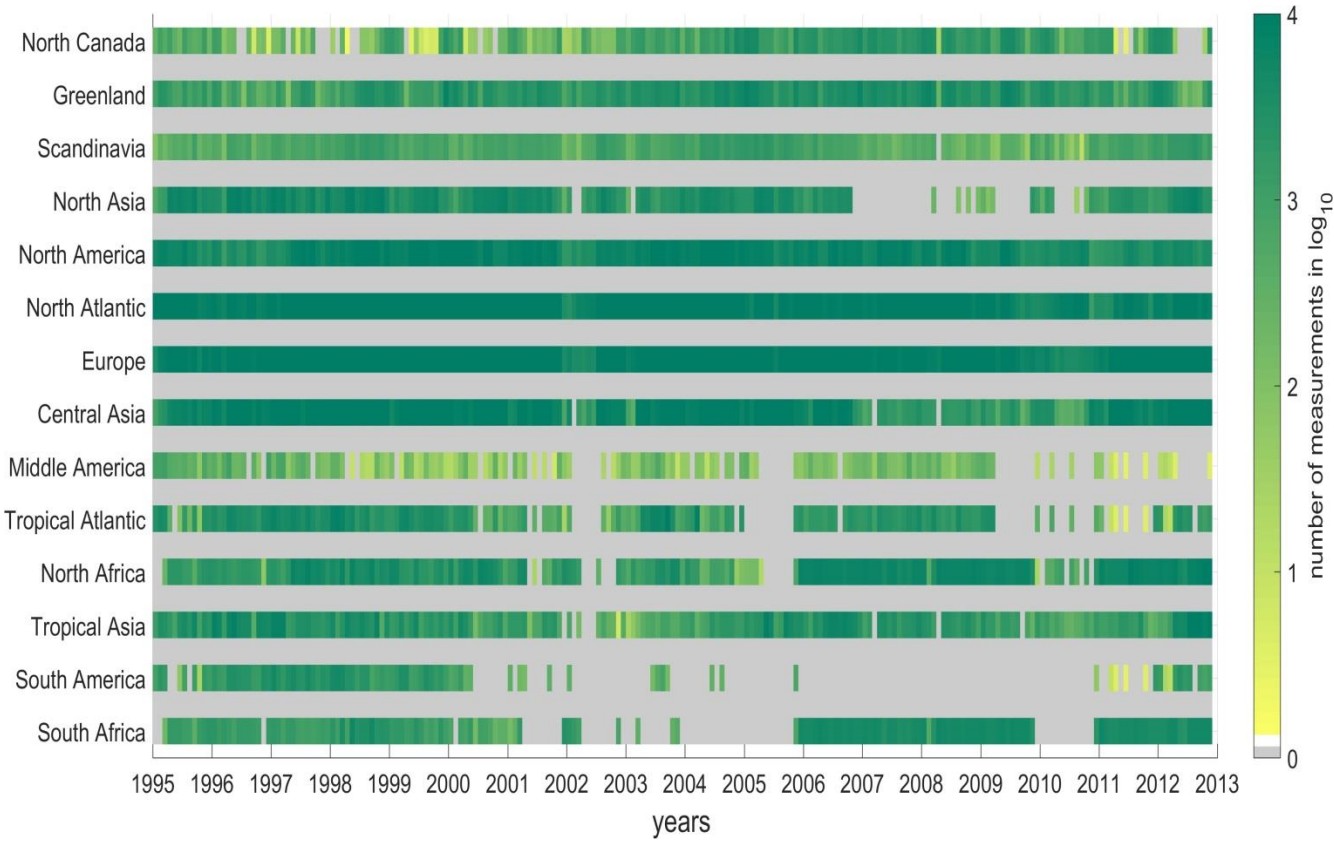

**Figure 4: Data density at cruise altitude (p < 350 hPa) for all defined regions in a logarithmic scale (color) for the analyzed period.**

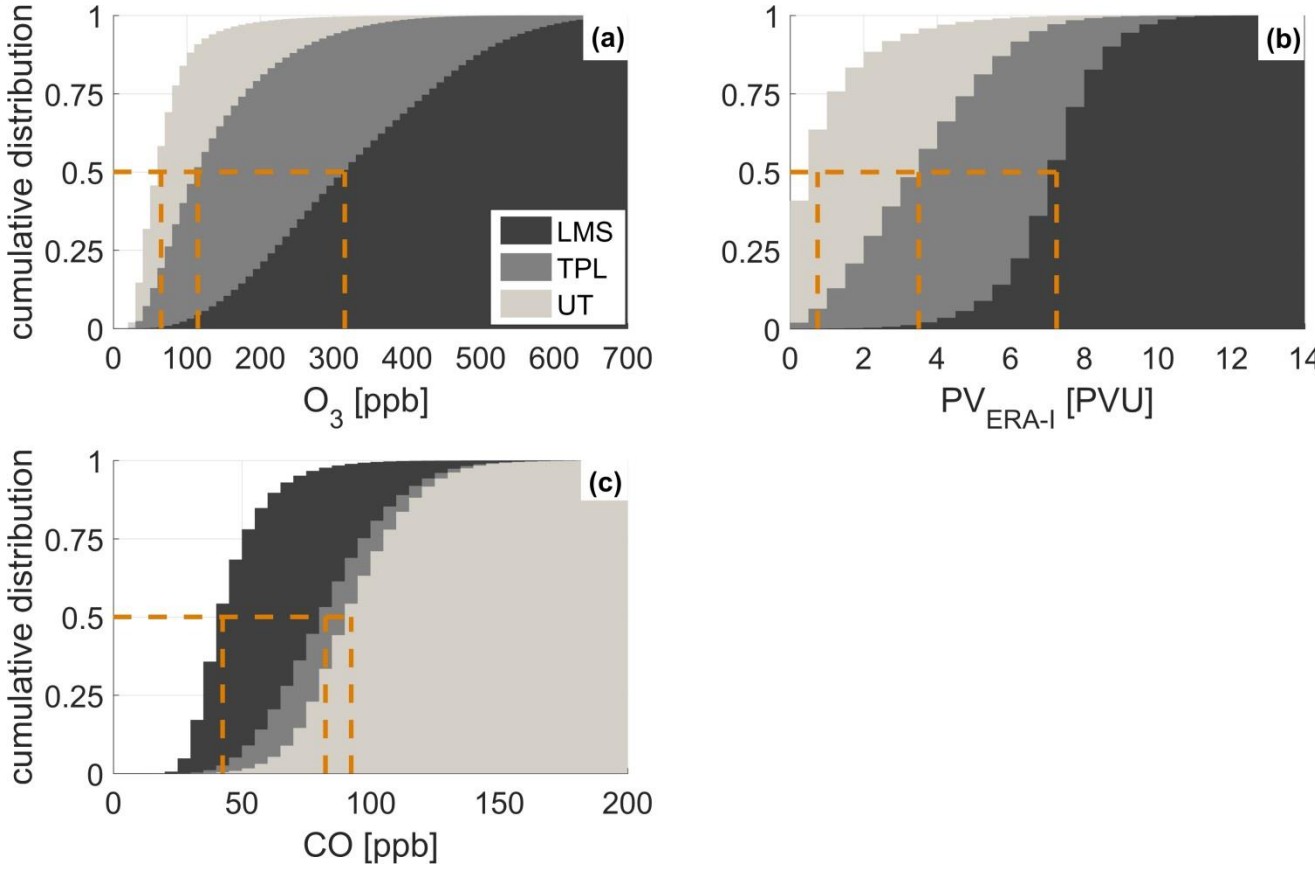

**Figure 5: Cumulative distribution of a) ozone, b) PV from ERA-I, and c) CO for the different vertical layers (LMS (dark), TPL (grey), UT (light grey)) over the North Atlantic region from January 1995 to December 2012. The dashed lines mark the values of each species when the cumulative distribution reached 50%.**

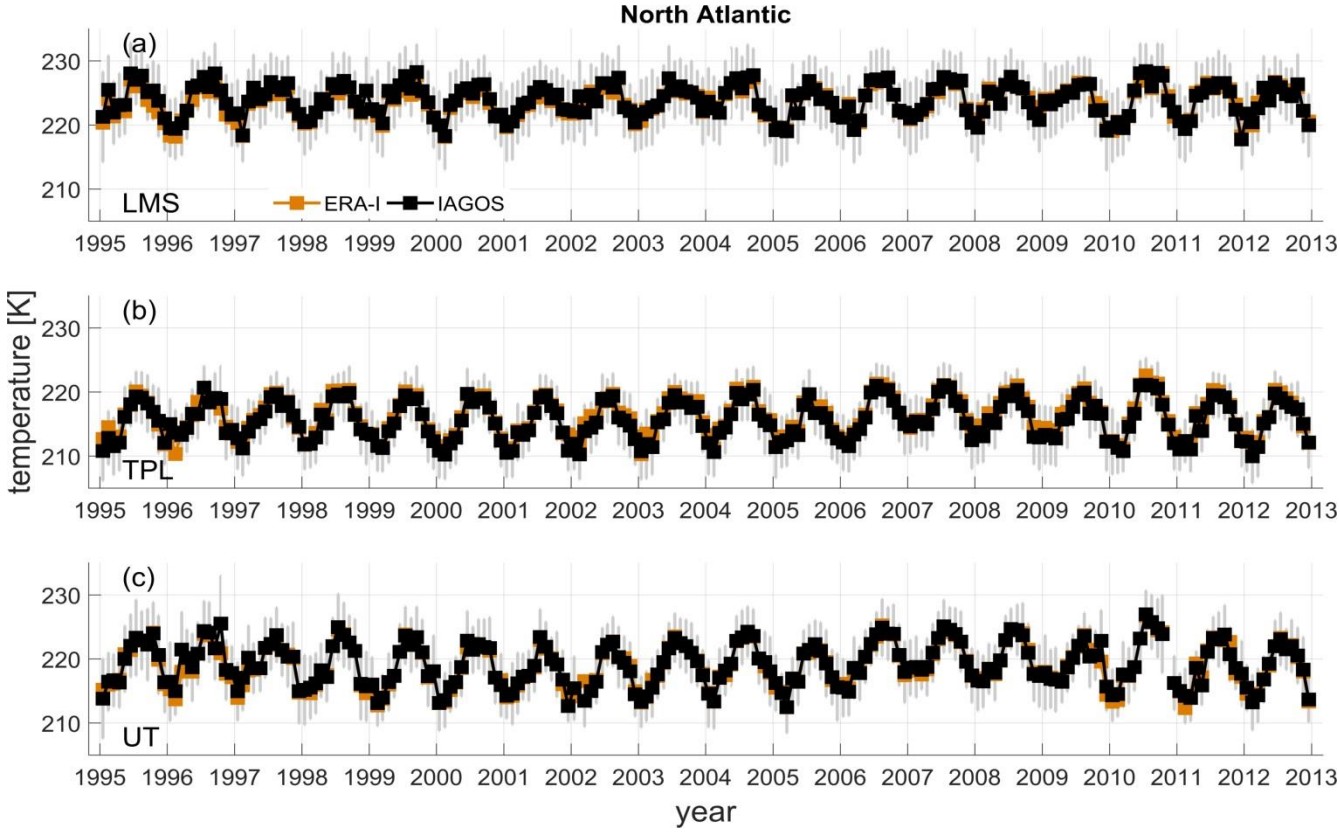

**Figure 6: Temperature time series of IAGOS observations (black) and ERA-I (orange) for the a) lowermost stratosphere, b) tropopause layer, and c) upper troposphere over the North Atlantic region. The grey lines show the standard deviation (1Ϭ) of the mean using the IAGOS observations for each month.**

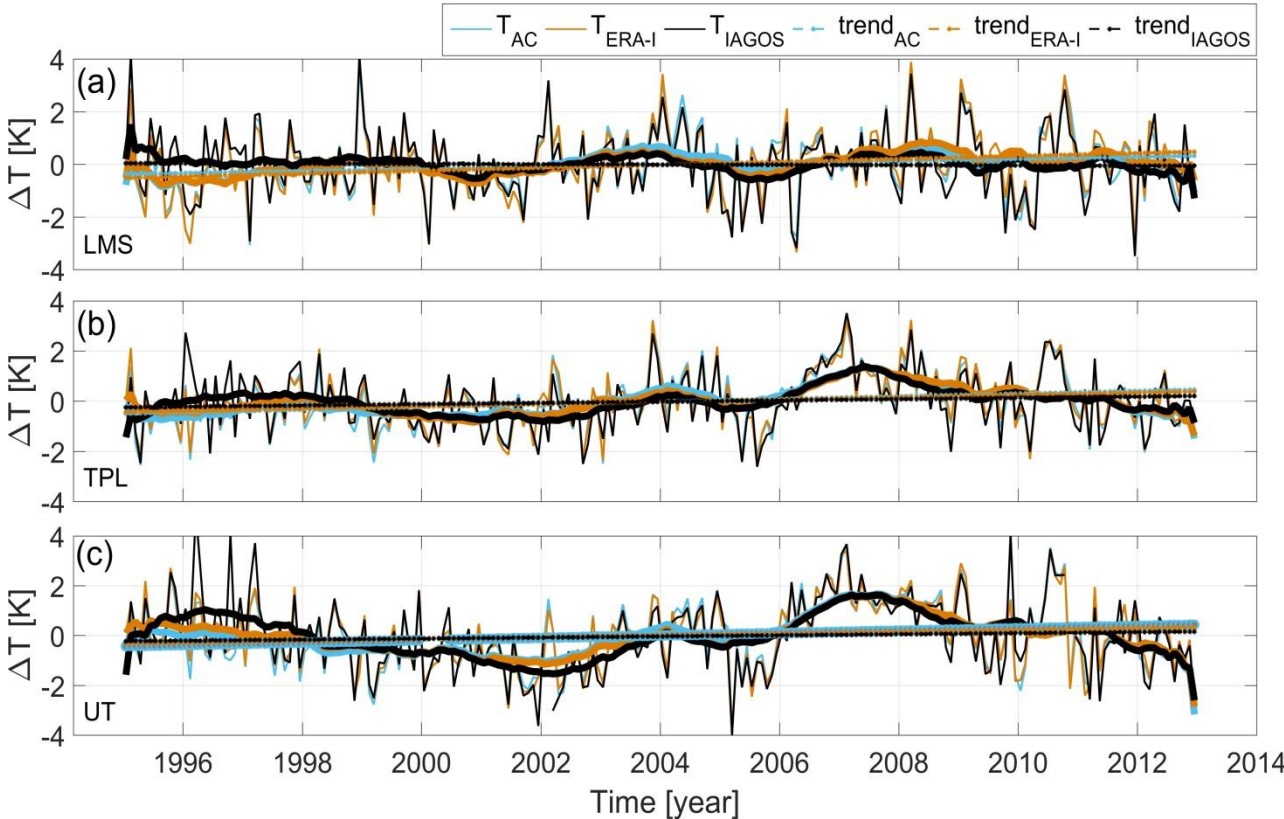

**Figure 7: Anomalies in monthly averaged temperature in the a) LMS b) TPL, and c) UT over the North Atlantic region from IAGOS observations (black), ERA-I (orange), and aircraft measurements (AC, light blue). The aircraft measurements are assumed to be comparable to AMDAR data. The anomalies are smoothed with a 12 month running mean to highlight the behavior of the time series, which reflects the temperature trends by linear regression analyses (dashed lines).**

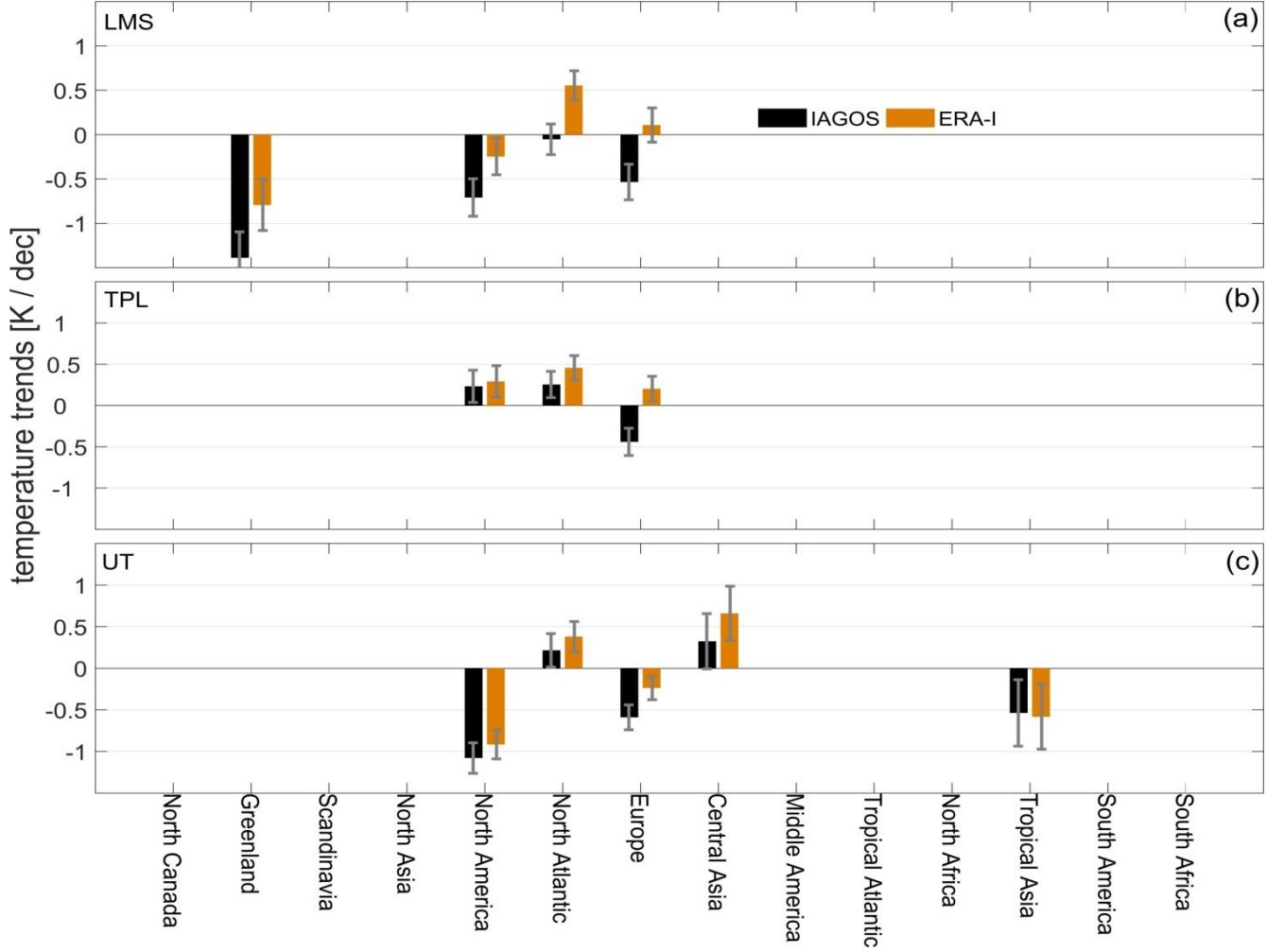

**Figure 8: Temperature trends over the analyzed period for each region obtained from the IAGOS observations and ERA-I for the a) LMS, b) TPL, and c) UT, where at least 90% of all months where available (see text for details).**

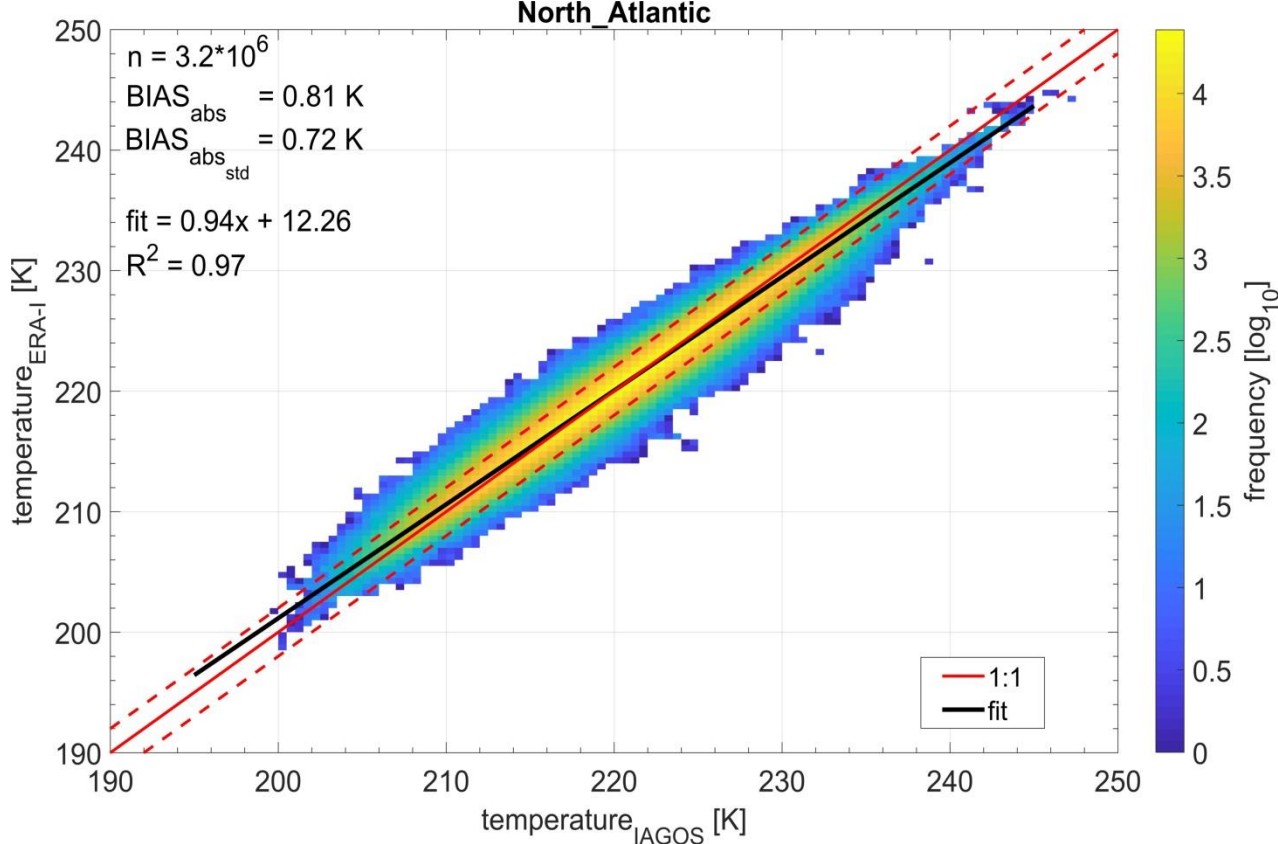

**Figure 9:** Temperature correlation from the observations and ERA-I over the North Atlantic region. The distribution is averaged in bin widths of 0.5 K and the color shows the number frequency with each bin in a logarithmic scale. The included numbers show the amount of data points, the absolute bias, the corresponding standard deviation and the linear robust regression fit (black line). The 1:1 line is shown in red, and its variation with 2 K as red dashes lines. The statistical values for all other regions are summarized in Tab. 3.

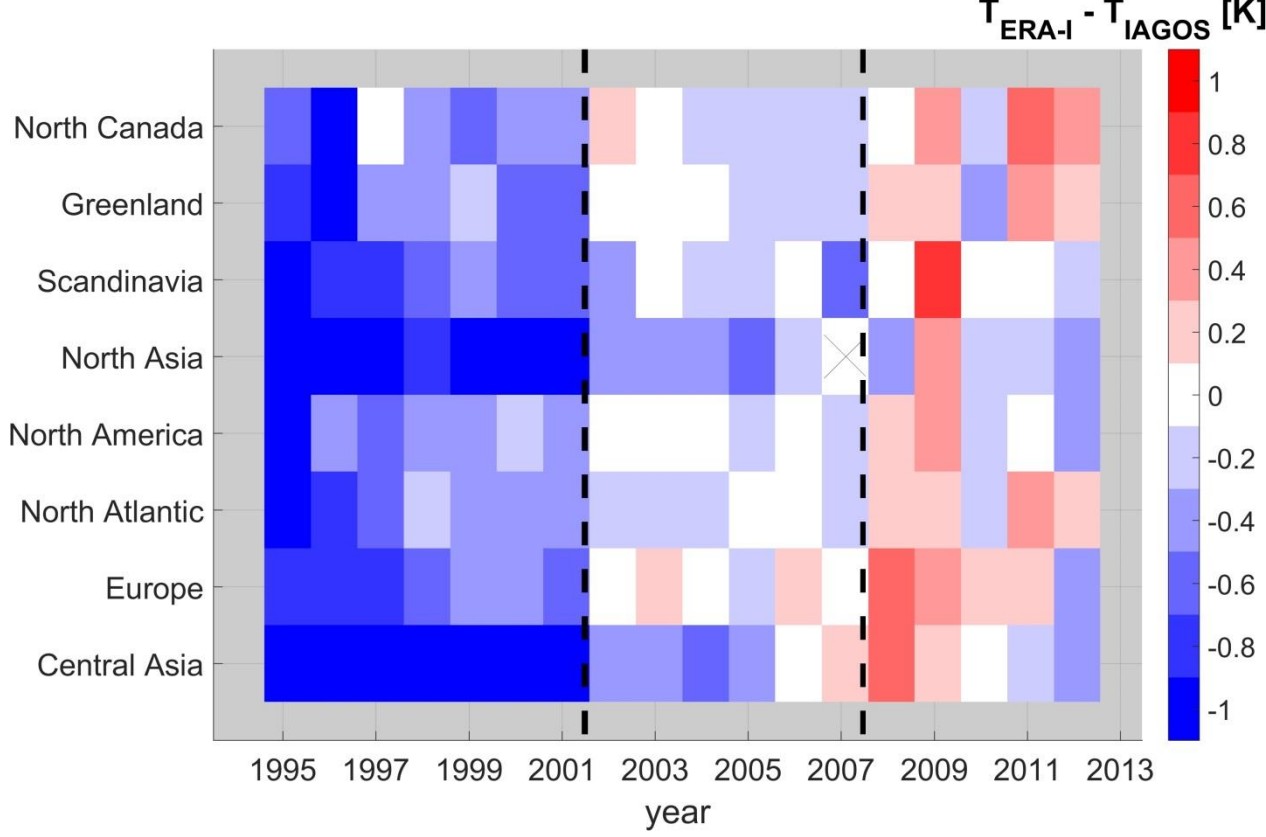

**Figure 10: Annual mean of the monthly mean difference between the observations and ERA-I for different region in the Northern Hemisphere in the lowermost stratosphere from 1995 to 2014. The warm colors corresponds to colder temperatures of the observations and blueish colors to warmer IAGOS temperatures compared to ERA-I. The dashed lines show clear break-points within the time series. The cross marks a year when the annual mean could not be calculated.**