# Peer review of "Two decades of in-situ temperature measurements in the upper troposphere and lowermost stratosphere from IAGOS long-term routine observation"

_Atmospheric Chemistry and Physics, 2017_

## Referee Comment (RC1) · Anonymous Referee #1 · 22 Jul 2017

This study presents a very valuable data set of 18 years of IAGOS temperature measurements in the UTLS region. Trend estimates are derived and compared with ERA-Interim reanalyses. The results are relevant and novel, and the paper is generally well written. However, a few very important aspects of the study are not clear and require clarification. I therefore recommend to accept the paper subject to major revisions.

Major comments:

A) An 18-year time period is short for trend analyses. I still think that the trend calcula-

tions in this study are useful, and I very much appreciate the efforts of the author team to produce such a long high-quality data set, but the time period issue must be discussed in the paper. I would like to see two additional trend calculations: (i) for IAGOS, how do the trends change if you skip the first or last year of your 18-year time series? Such a "sensitivity test" could be interesting to assess the robustness of trend values from 18 years compared to 17 years. The additional values could be included in Fig. 8. (ii) for ERA-Interim, you could compare the trends for the 18-year period with the full ERA-Interim period. Again, this would tell us something about how trends depend on the duration of the time period considered. A brief discussion of this should also be included in the concluding section.

B) I appreciate the efforts in data calibration, but I find it a bit disappointing that IAGOS data is only available until December 2012. Adding some of the recent years would also help with the issue mentioned above. Is there no way how you could include a few more years?

C) Unfortunately, I don't understand the method to distinguish between LMS, TPL and UT. I understand how you determine the pressure of the thermal TP from ERA-Interim; so the TPL is a 30-hPa deep tropopause-following layer, which varies in space and time (is this correct)? Then I am lost what "max(TPL)" and "min(TPL)" mean on p. 5: max and min over what? time or space? and how to you measure max/min? does it refer to pressure? It seems to me as if TPL is tropopause-following, but LMS and UT have fixed horizontal bounds, I find this very confusing.

D) p. 6 lines 13-15: It is very important whether IAGOS data has been assimilated in ERA-Interim, or not. This is not clear from the text. The first sentence says that "aircraft and other" data are assimilated in ERA-Interim, it seems that this does not include IAGOS. The next sentence then says "Note that IAGOS ... observations are not assimilated in any other NWP model ..." which sounds as if IAGOS is assimilated only in ERA-Interim, but I assume it is not assimilated by any reanalysis system. Then the "other" would be very misleading. Please clarify.

Minor comments:

1) p. 1 line 21: "temperature bias between observation and model data" sounds strange to me; do you mean the bias of the observations or the bias of the "model data" (note that reanalyses are not really just model data) or do you mean that both have a bias but that the biases differ?

2) p. 1 line 28: delete ";" after reference to Seidel et al., 2016

3) p. 2 line 7: "suffers" –> "suffer"

4) p. 2 line 15: first introduce abbreviation for LMS

5) p. 2 line 29: "Petersen et al., 2015" should read "Petersen, 2016" (single author, and paper appeared one year later)

6) p. 2 line 30: "including" is strange here, do you mean "assimilating"?

7) p. 2 line 31: "Drue" –> "Drüe"

8) p. 3 line 12: not clear what is meant by "both types of profile measurements"

9) p. 4 line 22: most readers are not familiar with the AIRTOSS-ICE campaign. What "research aircraft" has been used, and what type of temperature sensor?

10) p. 4 line 31: I understand "air temperature" but what is "total air temperature"?

11) p. 5 line 3: "maintained" sounds strange, maybe "made"?

12) p. 5 line 4: I find this a strange remark, it sounds as if the DWD model was used as a reference to test the accuracy of the observations ... maybe delete this sentence.

13) p. 5 line 21 and in other places: use "p" as symbol for pressure, not "P"

14) p. 5 line 30: it is fine that you use here the thermal tropopause, but the long list of chemical, ... tropopauses does not help the reader and is not relevant here. I suggest to delete the last 3.5 lines on this page.

[Figure]

15) p. 6 line 2: of course the Coriolis parameter is defined near the equator, but it is zero! Therefore PV goes to zero and the 2-pvu surface is not defined.

16) p. 6 line 19: Reichler et al. 2003 is not in the list of references

17) p. 6 line 21: this statement is very strange, the time resolution of ERA-Interim is 6 hours, so it cannot be reduced to 1 minute. Please explain how you interpolated 1-min values from ERA-Interim, by linear interpolation in time?

18) p. 8 line 3: "it is assumed ...": do you have a reference for this?

19) p. 8 line 6: "in this region" appears twice in the same sentence

20) p. 8 line 12: what do you mean by "local tropopause", maybe delete "local"?

21) p. 8 line 30: "where" –> "when"

22) p. 9 line 21: "then" –> "than"

23) p. 9 line 26: "annual averages of the monthly deviation": sounds complicated to me and it should be the same as just "annual deviation"?

24) p. 10 line 16: references should be in chronological order

25) p. 10 line 19: I don't understand this sentence: what aircraft measurements are you using here? IAGOS or AMDAR, and what is assumed to be similar?

26) p. 11 line 20: what is meant by "use IAGOS observations as anchor point"? Do you suggest to calibrate AMDAR data with IAGOS? And why do you not suggest that IAGOS observations should be assimilated in, e.g., ERA-5?

27) p. 15 line 13: I did not find a reference to Kuo et al. in the text

28) p. 16 line 9: volume and page numbers are missing

---

## Referee Comment (RC2) · Anonymous Referee #2 · 28 Jul 2017

This paper analyzes passenger-aircraft IAGOS temperature observations in the UTLS for the period 1995-2012, focusing on long-term trends. Comparisons with the ERA-Interim reanalysis data are also made, and change points in ERA-Interim are identified and discussed. Very careful data analysis has been made, and the discussion on ERA-Interim is very interesting. I think the manuscript can be published in ACP after considering the following comments.

1. As the authors explain in Introduction, the aircraft AMDAR data are known to have warm biases. I would like the authors to add some more explanations why the IAGOS

data are considered to be without such biases. Specifically, please consider (1) to add a similar figure to Figure 1 for the automated temperature instrument and (2) to add some technical descriptions for it in the last paragraph of Section 2.2. Is the temperature sensor material same? Is the instrument housing same? Is the correction method for adiabatic compression same? Are there other data processing procedures for both?

2. Please also describe the instruments for positions, pressure, and aircraft and wind speed/direction? The pressure measurement uncertainty may be important for analyzing temperature trends because there are temperature gradients in the UTLS so that pressure errors would result in slightly different temperature values. Aircraft/wind speed/direction might be important for the correction of adiabatic compression (or not?).

3. I have some questions related to the data analysis methods. (Please note that I am not asking the authors to re-do the data analysis.)

3-1. Are there any sampling biases/trends in vertical within each layer? Because there are temperature gradients in the UTLS, if the flight level would have changed systematically over the period, we would get apparent temperature trends.

3-2. Similarly, are there any sampling trends in horizontal? My understanding is that the pilots are trying to avoid turbulences i.e., the regions with strong (horizontal) shears. If the westerly jets have some trends in their locations (cf. Davis and Rosenlof, 2012; Williams, 2017), the aircraft paths may have some systematic changes that could give apparent temperature trends.

3-3. For comparison with ERA-Interim, another way could be to pick up 6 hourly data (i.e., at the times when the reanalysis data are provided) from the IAGOS data. Linear interpolations to 1 min from 6 hourly data may produce unnecessary spreads in the difference of the two data.

4. Page 7, lines 23-24 and Figure 7. The warm phase during 2006-2008 might be

related to a volcanic eruption (the Soufrière Hills at 16N on 20 May 2006). See Vernier et al. (2011).

References:

Sean M. Davis, S. M., K. H. Rosenlof (2012), A Multidiagnostic Intercomparison of Tropical-Width Time Series Using Reanalyses and Satellite Observations, J. Clim., 25, 1061-1078, https://doi.org/10.1175/JCLI-D-11-00127.1.

Vernier, J.-P., et al. (2011), Major influence of tropical volcanic eruptions on the stratospheric aerosol layer during the last decade, Geophys. Res. Lett., 38, L12807, doi:10.1029/2011GL047563.

Williams, P.D., Increased light, moderate, and severe clear-air turbulence in response to climate change, Adv. Atmos. Sci. (2017) 34: 576. https://doi.org/10.1007/s00376-017-6268-2.

---

## Author Comment (AC1) · 5 Sep 2017

The comment was uploaded in the form of a supplement:
https://www.atmos-chem-phys-discuss.net/acp-2017-494/acp-2017-494-AC1-supplement.zip
* * *

---

## Author Response (AR1)

[revised manuscript text omitted]

temperature measurement of the research aircraft are regular quality controlled and are well calibrated. Figure 2Figure 2 shows the temperature correlation and the temperature bias (ΔT) at pressures below 400 hPa during seven flights. The general behavior between both temperature measurements agreed well along the flight tracks. The mean deviation is ΔT = - 0.3 K, with a pressure dependency to a smaller temperature bias towards 200 hPa. The temperature correlation is high and the temperature bias is smaller than the overall uncertainty of 0.5 K (sensor and adiabatic compression correction), which demonstrates the capability of the IAGOS temperature sensor to measure the ambient temperature at cruise altitude very precisely.

The temperature measured from the aircraft ($T_{AC}$) is based on total air temperature (TAT) designed for subsonic aircraft (Goodrich Corporation, formerly Rosemount Aerospace). The total air temperature is defined as the ambient air temperature plus the temperature increase due to adiabatic compression in the Rosemount housing. Typically three TAT-sensors (platinum resistance sensor) are installed on at the nose region of the aircraft, but in general only one is used for the pilots and stored for IAGOS. The other two sensors are used to monitor the differences between all TAT-sensors. In general, the airline follow the AMDAR quality recommendations (WMO, 2003), and the TAT-sensors are regular checked by visible inspection. An exchange of one TAT-sensor is performedmaintained, if it differs more than 3°C from the other two TAT-sensors. For the Lufthansa fleet, $T_{AC}$ is compared additionally to models from the German Weather Service to ensure an accuracy of ±3°C (pers. Communication, Lufthansa Technik AG).

[revised manuscript text omitted]

Reply to
This study presents a very valuable data set of 18 years of IAGOS temperature measurements in the UTLS region. Trend estimates are derived and compared with ERA-Interim reanalyses. The results are relevant and novel, and the paper is generally well written. However, a few very important aspects of the study are not clear and require clarification. I therefore recommend to accept the paper subject to major revisions.

**We thank the referee for her/his comments, which we address (in bold) point by point in our reply below.**

Major comments:

A) An 18-year time period is short for trend analyses. I still think that the trend calculations in this study are useful, and I very much appreciate the efforts of the author team to produce such a long high-quality data set, but the time period issue must be discussed in the paper. I would like to see two additional trend calculations: (i) for IAGOS, how do the trends change if you skip the first or last year of your 18-year time series? Such a "sensitivity test" could be interesting to assess the robustness of trend values from 18 years compared to 17 years. The additional values could be included in Fig.8.

**Answer: The robustness of the trend analysis was tested with the Mann-Kendall test using different lengths of the time series. In order to further clarify this point, we have now added one sentence in section 3.2 and updated table S4 in the supplement:**

**"The robustness of temperature trends was tested by skipping the first or last year of the 18-year period. Within all layers and all regions each trend keeps the same sign and the trend values varied within the standard error. The only exception was the upper troposphere over North America where the temperature trend changed from slight positive trend (18 years) to neutral when the final year was removed. "**

(ii) for ERA-Interim, you could compare the trends for the 18-year period with the full ERA-Interim period. Again, this would tell us something about how trends depend on the duration of the time period considered. A brief discussion of this should also be included in the concluding section.

**Answer: We agree that the full time period would give additional information about the behavior of the ERA-Interim data, but the ERA-I temperature data is interpolated along the flight tracks, therefore it covers only the IAGOS period. We could create artificial flight tracks for other years, however the focus of this study is to present the IAGOS temperature measurements and not to evaluate ERA-I.**

B) I appreciate the efforts in data calibration, but I find it a bit disappointing that IAGOS data is only available until December 2012. Adding some of the recent years would also help with the issue mentioned above. Is there no way how you could include a few more years?

**Answer: Between 2011 and 2014 some MOZAIC aircraft were still operating. The first IAGOS aircraft started to measure in July 2011 and the second in June 2012. The IAGOS temperature and humidity data validation tool had to be re-developed and is currently under evaluation for measurements past 2013. The data coverage of the MOZAIC aircrafts for the period past 2013 is too sparse for a meaningful extension of our trend analysis. Therefore we limited our data analysis to the period until 2012.**

C) Unfortunately, I don't understand the method to distinguish between LMS, TPL and UT. I understand how you determine the pressure of the thermal TP from ERA-Interim; so the TPL is a 30-hPa deep tropopause-following layer, which varies in space and time (is this correct)? Then I am lost what "max(TPL)" and "min(TPL)" mean on p. 5: max and min over what? time or space? and how to you measure max/min? does it refer to pressure? It seems to me as if TPL is tropopause-following, but LMS and UT have fixed horizontal bounds, I find this very confusing.

**Answer: We are sorry for the confusion. TPL is related to the range of the pressure altitude of the tropopause +-15 hPa. Max/min is related to the upper/lower limit of this range around the tropopause. To clarify, all three layers are tropopause-following and we improved the layer definition in the manuscript:**

**LMS : $p < p_{TPHWMO}$ - 15 hPa, which is limited by the maximum cruise altitude (p ~ 190 hPa)**

**TPL: $p = p_{TPHWMO} \pm 15$ hPa**

**UT: $p > p_{TPHWMO} + 15$ hPa, limited to 350 hPa**

D) p. 6 lines 13-15: It is very important whether IAGOS data has been assimilated in ERA-Interim, or not. This is not clear from the text. The first sentence says that "aircraft and other" data are assimilated in ERA-Interim, it seems that this does not include IAGOS. The next sentence then says "Note that IAGOS ... observations are not assimilated in any other NWP model ..." which sounds as if IAGOS is assimilated only in ERA-Interim, but I assume it is not assimilated by any reanalysis system. Then the "other" would be very misleading. Please clarify.

**Answer: IAGOS temperature measurements are not assimilated into ERA-Interim or any other model. We deleted the word "other" and apologize for the confusion.**

Minor comments:

1) p. 1 line 21: "temperature bias between observation and model data" sounds strange to me; do you mean the bias of the observations or the bias of the "model data" (note that reanalyses are not really just model data) or do you mean that both have a bias but that the biases differ?

**Answer: We agree with the reviewer that it is not very clear. We exchanged the word "bias" by "difference".**

2) p. 1 line 28: delete ";" after reference to Seidel et al., 2016

**Answer: Done**

3) p. 2 line 7: "suffers" –> "suffer"

**Answer: Changed**

4) p. 2 line 15: first introduce abbreviation for LMS

**Answer: Done**

5) p. 2 line 29: "Petersen et al., 2015" should read "Petersen, 2016" (single author, and paper appeared one year later)

**Answer: Sorry for the mistake, the reference is updated.**

6) p. 2 line 30: "including" is strange here, do you mean "assimilating"?

**Answer: Changed**

7) p. 2 line 31: "Drue" –> "Drüe"

**Answer: Changed**

8) p. 3 line 12: not clear what is meant by "both types of profile measurements"

**Answer: Should read: *"… with temperature profiles from in-situ measurements and from the model…"***

9) p. 4 line 22: most readers are not familiar with the AIRTOSS-ICE campaign. What "research aircraft" has been used, and what type of temperature sensor?

**Answer: We included the information in the text**

*"During this campaign, the IAGOS temperature instrument was installed on the research aircraft (Learjet 35A), and provides the opportunity to compare both temperature measurements. The aircraft temperature measurement was made with Pt-100 thermistor mounted in the same type of Rosemount as for the IAGOS temperature measurements. The temperature sensor of the research aircraft has been calibrated regularly with an uncertainty of about 0.5 to 1.0 K."*

10) p. 4 line 31: I understand "air temperature" but what is "total air temperature"?

**Answer: We included into the text:**

*"The total air temperature is defined as the ambient air temperature plus the temperature increase due to adiabatic compression in the Rosemount housing."*

11) p. 5 line 3: "maintained" sounds strange, maybe "made"?

**Answer: We changed *"maintained"* to *"performed"*.**

12) p. 5 line 4: I find this a strange remark, it sounds as if the DWD model was used as a reference to test the accuracy of the observations ... maybe delete this sentence.

**Answer: We agree with the reviewer that the sentence leads to misunderstanding and deleted it.**

13) p. 5 line 21 and in other places: use "p" as symbol for pressure, not "P"

**Answer: Changed**

14) p. 5 line 30: it is fine that you use here the thermal tropopause, but the long list of chemical,…tropopauses does not help the reader and is not relevant here. I suggest to delete the last 3.5 lines on this page.

**Answer: We follow the suggestion of the reviewer and delete the different tropopause definitions.**

15) p. 6 line 2: of course the Coriolis parameter is defined near the equator, but it is zero! Therefore PV goes to zero and the 2-pvu surface is not defined.

**Answer: Thanks for the correction. We changed the sentence to:**

*The thermal tropopause is valid within all latitude bands, whereas the dynamical tropopause cannot be used in tropical regions, because the Coriolis parameter is zero, therefore PV goes to zero and the 2-pvu surface is not defined (Boothe and Homeyer, 2017).*

16) p. 6 line 19: Reichler et al. 2003 is not in the list of references

**Answer: Sorry for the mistake. It is now included.**

17) p. 6 line 21: this statement is very strange, the time resolution of ERA-Interim is 6 hours, so it cannot be reduced to 1 minute. Please explain how you interpolated 1-min values from ERA-Interim, by linear interpolation in time?

**Answer: The ERA-I data is linearly interpolated (time and space) along the flight path in 4 s temporal resolution equivalent to the in-situ measurements. Afterwards the in-situ data and the ERA-I data were averaged to 1 min to reduce the size of data as written on p. 4 line: 19.**

18) p. 8 line 3: "it is assumed ...": do you have a reference for this?

**Answer: We included the reference *"Kandhu et al. 2016"* at the end of the sentence:**

**Khandu, Awange, J. L. and Forootan, E.: Interannual variability of temperature in the UTLS region over Ganges–Brahmaputra–Meghna river basin based on COSMIC GNSS RO data, Atmos. Meas. Tech., 9(4), 1685–1699, doi:10.5194/amt-9-1685-2016, 2016.**

19) p. 8 line 6: "in this region" appears twice in the same sentence

**Answer: We rephrased the sentence.**

*"Another reason could be related to a higher variability of the temperature due to large scale influence or simply that the data coverage is still too poor in this region, which might lead to a higher variability of the local temperatures which then mask the temperature trend."*

20) p. 8 line 12: what do you mean by "local tropopause", maybe delete "local"?

**Answer: We agree with the reviewer and deleted local.**

21) p. 8 line 30: "where" –> "when"

**Answer: Changed**

22) p. 9 line 21: "then" –> "than"

**Answer: Changed**

23) p. 9 line 26: "annual averages of the monthly deviation": sounds complicated to me and it should be the same as just "annual deviation"?

**Answer: Changed as suggested**

24) p. 10 line 16: references should be in chronological order

**Answer: Corrected**

25) p. 10 line 19: I don't understand this sentence: what aircraft measurements are you using here? IAGOS or AMDAR, and what is assumed to be similar?

**Answer: We demonstrated that using only the aircraft temperature measurements ("AMDAR like") the aircraft temperature trend follow mostly the ERA-Interim temperature trend, which is an indication that here is a positive bias in the aircraft data. However, the aircraft temperature data were not validated after the AMDAR recommendations, therefore we wrote: "…, which we assume to be comparable to AMDAR measurements". We included the "$(T_{AC})$" to be more specific.**

26) p. 11 line 20: what is meant by "use IAGOS observations as anchor point"? Do you suggest to calibrate AMDAR data with IAGOS? And why do you not suggest that IAGOS observations should be assimilated in, e.g., ERA-5?

**Answer: The amount of IAGOS temperature measurements are much less compared to the amount of AMDAR temperature measurements and the benefits would be minor in respect to additional measurements. However, both temperature measurements (AMDAR and IAGOS) could be assimilated into ERA5, than IAGOS data could be uses as anchor point for the bias correction of the AMDAR data.**

27) p. 15 line 13: I did not find a reference to Kuo et al. in the text

**Answer: Sorry for the mistake. The reference is an artefact from an earlier version of the manuscript.**

28) p. 16 line 9: volume and page numbers are missing

**Answer: Fixed**

Table S4: Temperature trends of ERA-I and from the IAGOS observations within the LMS, TPL and UT as shown in table3 and compared to temperature trends derived from 17 years skipping the first (light gray) or last year (dark gray) in the analyses.

| Region | ERA-I | | | | IAGOS | | | |
|---|---|---|---|---|---|---|---|---|
| | $\Delta T_{18yr}$ K/dec | SE K/dec | $\Delta T_{17yr, first}$ K/dec | $\Delta T_{17yr, last}$ K/dec | $\Delta T_{18yr}$ K/dec | SE K/dec | $\Delta T_{17yr, first}$ K/dec | $\Delta T_{17yr, last}$ K/dec |
| *LMS* | | | | | | | | |
| Greenland | -0.79 | 0.29 | -1.03 | -0.83 | -1.39 | 0.29 | -1.48 | -1.45 |
| North America | -0.25 | 0.21 | -0.38 | -0.21 | -0.71 | 0.21 | -0.73 | -0.76 |
| North Atlantic | +0.56 | 0.17 | +0.53 | +0.59 | -0.05 | 0.17 | +0.02 | -0.01 |
| Europe | +0.11 | 0.19 | +0.12 | +0.14 | -0.53 | 0.20 | -0.49 | -0.62 |
| *TPL* | | | | | | | | |
| North America | +0.29 | 0.19 | +0.11 | +0.45 | +0.23 | 0.20 | -0.02 | +0.42 |
| North Atlantic | +0.46 | 0.15 | +0.42 | +0.62 | +0.25 | 0.16 | +0.20 | +0.38 |
| Europe | +0.20 | 0.15 | +0.17 | +0.19 | -0.44 | 0.17 | -0.46 | -0.57 |
| *UT* | | | | | | | | |
| North America | -0.92 | 0.17 | -0.99 | -0.82 | -1.08 | 0.18 | -1.16 | -1.00 |
| North Atlantic | +0.38 | 0.18 | +0.52 | +0.58 | +0.22 | 0.20 | +0.33 | +0.43 |
| Europe | -0.24 | 0.14 | -0.21 | -0.29 | -0.59 | 0.15 | -0.55 | -0.71 |
| Central Asia | +0.66 | 0.33 | +0.55 | +0.91 | +0.32 | 0.33 | +0.27 | +0.54 |
| Tropical Asia | -0.58 | 0.39 | -0.43 | -0.24 | -0.54 | 0.04 | -0.30 | -0.21 |

Reply to
This paper analyzes passenger-aircraft IAGOS temperature observations in the UTLS for the period 1995-2012, focusing on long-term trends. Comparisons with the ERA-Interim reanalysis data are also made, and change points in ERA-Interim are identified and discussed. Very careful data analysis has been made, and the discussion on ERA-Interim is very interesting. I think the manuscript can be published in ACP after considering the following comments.

**We thank the referee for her/his comments, which we address (in bold) point by point in our reply below.**

1. As the authors explain in Introduction, the aircraft AMDAR data are known to have warm biases. I would like the authors to add some more explanations why the IAGOS data are considered to be without such biases. Specifically, please consider

(1) to add a similar figure to Figure 1 for the automated temperature instrument and

Answer: **The aircraft temperature sensor position and Rosemount housing are located at three different spots at the nose region of the aircraft (may differ from airline to airline), but the system is equivalent to the IAGOS system and this is shown.**

(2) to add some technical descriptions for it in the last paragraph of Section 2.2. Is the temperature sensor material same? Is the instrument housing same? Is the correction method for adiabatic compression same? Are there other data processing procedures for both?

Answer: **Moninger et al. (2003) summarized that the aircraft temperature sensor can be affected by moisture, and dirt (insects and other materials), which can lead to a coating of the probe and to a drift of the sensor signal. The sensor type is also a platinum resistance sensor, as used in IAGOS. Temperature measurement uncertainties increase due to the adiabatic temperature correction, which includes the uncertainties of the static pressure and Mach number (calculated from the aircraft speed). The aircraft temperature correction of the adiabatic compression is calculated by the avionics of the aircraft itself. The entire procedures and uncertainty determination are given in Stickney et al., 1994 and Helten et al., 1998.  The aircraft temperature is provided from avionics of the aircraft with a precision of +/- 0.5 K and maximum uncertainty of +/- 2 K based on AMDAR quality criteria (WMO, 2003). Please note, we wrote in the introduction of the manuscript: "** However, Ballish and Kumar (2008) and Drüe et al. (2008) identified that the AMDAR aircraft temperature is strongly affected by a warm bias, which can fluctuate by altitude, aircraft type and phase of flight, while the reason for this bias is not fully understood (Ingleby et al., 2016)."

IAGOS changes regularly (currently 2-3 months interval) the temperature/humidity sensors to avoid drifts of the signal (e.g. due to dirt) and the quality of each sensor is checked before and after each deployment in the laboratory. Also for each 4-s data point we provide a quality flag and the uncertainty is calculated, which is not the case for the aircraft temperature data. As far as we know, AMDAR temperature measurements aren't considered if they differ more than 2 K from the NWPs. So we think that we might be able to avoid a warm temperature bias, because we have a more frequent control and exchange of the temperature sensor.

**We included additional information in the last paragraph of Sec 2.2 in the revised version.**

2. Please also describe the instruments for positions, pressure, and aircraft and wind speed/direction? The pressure measurement uncertainty may be important for analyzing temperature trends because there are temperature gradients in the UTLS so that pressure errors would result in slightly different temperature values. Aircraft/wind speed/direction might be important for the correction of adiabatic compression (or not?).

**Answer:** **The instruments might vary for each aircraft type, a general overview about the uncertainty of the aircraft data within the IAGOS project was given by Petzold et al., 2015. It is true that the aircraft speed is crucial for the correction of the adiabatic compression, which is applied for each individual aircraft based on the in-situ measurements. The correction factor is calculated using the following equation:**

**$T_{total}$ / $T_{ambient}$= (1 + (γ-1)/2) $M^2$)**

**With $T_{total}$ as the total air temperature measured from the sensor. γ is the ratio of the specific heats of air at constant pressure and volume, respectively. M is the Mach number which is calculated using the aircraft speed divided by speed of sound in air. The total uncertainty of the temperature measurements is calculated with 0.5 K and includes the temperature sensor uncertainty of 0.25 K plus the uncertainty for the correction of the adiabatic compression (Helten et al., 1998). At cruise altitude, the Mach number is rather constant at M= 0.81+-0.01. More details are given in the standard operation procedure (SOP) of the IAGOS capacitive Hygrometer (ICH), available at www.iagos.org.**

**The aircraft pressure measurements are used to derive the aircraft position relative to the thermal tropopause calculated from ERA-Interim. The uncertainty from the pressure sensor (0.35 hPa) which is small compared to our range of the tropopause layer (+/- 15 hPa). Note, we wrote in the manuscript, that we evaluate the vertical position of the aircraft using in-situ measurements of CO, O3 and PV.**

**We added additional information at different parts in the manuscript.**

3. I have some questions related to the data analysis methods. (Please note that I am not asking the authors to re-do the data analysis.)

3-1. Are there any sampling biases/trends in vertical within each layer? Because there are temperature gradients in the UTLS, if the flight level would have changed systematically over the period, we would get apparent temperature trends.

3-2. Similarly, are there any sampling trends in horizontal? My understanding is that the pilots are trying to avoid turbulences i.e., the regions with strong (horizontal) shears. If the westerly jets have some trends in their locations (cf. Davis and Rosenlof, 2012; Williams, 2017), the aircraft paths may have some systematic changes that could give apparent temperature trends.

**Answer: We looked at the beginning in detail at the horizontal and vertical data coverage for each specific region, and we did not find a shift of the data coverage within each region over the study period. Over most regions the aircraft fly along distinct routes. Over the North Atlantic and Tropical Atlantic regions, the spread of the fight tracks is larger, which corresponds to flight routes with different destinations and avoiding severe weather conditions. However, we agree with the reviewer that it would be worth to look more into details of the aircraft measured wind fields to study changes of the wind pattern, which, however, is not the focus of this study.**

3-3. For comparison with ERA-Interim, another way could be to pick up 6 hourly data (i.e., at the times when the reanalysis data are provided) from the IAGOS data. Linear interpolations to 1 min from 6 hourly data may produce unnecessary spreads in the difference of the two data.

**Answer: The ERA-Interim data is interpolated along the flight path for every 4s and then averaged to 1 min to reduce the amount of data. If we compare point by point, the spread is large between the temperature of ERA-I and in-situ as we wrote in the manuscript and show in Fig 9. However over 93% of this spread is smaller than 2 K for the North Atlantic, which reflects the generally good agreement between these two data sets. Of course the IAGOS data could be interpolated on a model grid with the same time resolution, but from our point of view, this would lead to a new study with the focus on the evaluation the ERA-Interim model. For such study, it would be favorable to include additional data sets (e.g MERRA2, JRA-55 and others), which in fact is subject of ongoing work.**

4. Page 7, lines 23-24 and Figure 7. The warm phase during 2006-2008 might be related to a volcanic eruption (the Soufrière Hills at 16N on 20 May 2006). See Vernier et al. (2011).

**Answer: The reviewer is right that the mentioned warm phase could be related to the volcanic eruption. However, this hypothesis is speculative and we decided to avoid it and leave it for a future study.**